# Selecting Priority Policy Strategies for Sustainability of Micro, Small, and Medium Enterprises in Takalar Regency

**Syamsari Syamsari [1], Muhammad Ramaditya [2,\*], Irma Andriani [3] and Ayu Puspitasari [1]**

1   Institut Teknologi Pertanian, South Sulawesi, Makassar 92255, Indonesia
2   Sekolah Tinggi Ilmu Ekonomi Indonesia Jakarta, Jakarta Timur 13220, Indonesia
3   Department of Biology, Universitas Hassanudin, South Sulawesi, Makassar 90245, Indonesia
\*   Correspondence: ramaditya@stei.ac.id

**Abstract:** The current study aims to develop a strategic policy for the micro, small, and medium enterprises (MSMEs) resilience system to deal with disruption during uncertainty and maintain sustainability. Previous studies stated that the MSMEs mortality rate was high, specifically in the first five years. This was caused by various internal weaknesses and the inability to deal with disruption amid uncertainty. The enterprises need government intervention to strengthen their resilience. Therefore, this study aimed to design a government policy model for a resilience system that connects actors, factors, goals, and alternative strategies during uncertainty. Since disruptions are complex, a holistic approach comprising resource theory, entrepreneurial orientation, corporate entrepreneurship, and entrepreneurship ecosystem was used. The method used is the Fuzzy Analytical Hierarchy Process, involving 20 experts as respondents. The results showed that certain factors, actors, goals, and strategies must be prioritized in formulating a policy strategy for the sustainability system in Takalar Regency, Indonesia. The main factor is entrepreneurial orientation, while the actor is highly resilient MSMEs. Additionally, the goal is for the market to accept products, while innovative transformation is the alternative priority strategy to be implemented.

**Keywords:** entrepreneurship; Fuzzy AHP; sustainability; small and medium enterprises





## 1. Introduction

Uncertainty hinders or delays decision-making and action and arises due to high competition, open markets, and changes in the political and structural framework [1,2]. It is also caused by population growth, resource constraints, and climate change [3,4]. Organizations operate in an increasingly complex and volatile world full of unexpected events, such as the economic crises of 1997–1998 and 2007–2009, and the COVID-19 pandemic [5]. The incidents have caused instability and increased the risk to MSMEs' sustainability [6]. The sustainability of MSMEs should be protected because they are essential in economic development through job creation, investment, and increasing gross domestic product in almost all countries, including Indonesia [7]. However, this country's MSMEs are dominated by self-employment and traditional enterprises, have low productivity and product quality because they use a manual system and less technology, and generally serve small and local markets. These weaknesses make them unable to upgrade regarding trade liberalization, technological change, and increasing demand for high-quality products [8]. The main reason for the inability to upgrade is limited and low-quality human resources [9]. As a result, MSMEs have become vulnerable to uncertainty [10] because resources provide natural resilience [11].

The internal weaknesses and disruption during uncertainty have made MSMEs unable to build business sustainability independently, necessitating government help [12,13]. As part of the entrepreneurship ecosystem, government policies increase the ability to face disruption and maintain MSMEs' sustainability [14,15]. These policies include forecasting

and anticipating possible disruptions in the future [16]. The establishment of business resilience starts with planning and needs engineering from its inception [11]. MSMEs must be prepared to overcome more prominent and critical situations, rise from failure, and expand their business [17]. Entrepreneurs have been reactive in dealing with uncertainties because resilience is tested to be built following a disturbance, making businesses vulnerable to disruptions [18].

This study aimed to create a policy model for the MSMEs' resilience system to deal with disruption during uncertainty using a holistic approach comprising several theories [2]. The holistic approach combines resource and entrepreneurial theories that support business success, including entrepreneurial orientation, corporate entrepreneurship, and entrepreneurship ecosystem. The combination was used to develop a policy model for the resilience system to face disruption during uncertainty using the Fuzzy AHP method. The method resulted in actors, factors, objectives, and alternative strategies as the core of the government policy model.

Studies on resilience and sustainability in various research objects have been carried out using various methods. The resilience of the supply chain network in an uncertain environment is carried out by evaluating scenarios, namely, strategies and processes [19]. One of them is the Research in the fisheries sector to support food security uses descriptive qualitative analysis, design thinking, and lean business model canvas [19]. Resilience in the agricultural sector has been investigated using expert interviews, statistical analysis, and econometric modeling [19]. The Fuzzy AHP method has been used to test the components of supply chain resilience, namely robustness, agility, leanness, and flexibility [20].

Therefore, this study is essential because resilience in developing countries is a current-research focus [20]. Since studies on this subject are still developing, there needs to be more clarity regarding entrepreneurial resilience, its elements, and how it could be improved. A study on the resilience of tourism business entrepreneurs in Spain resulted in three resilience dimensions: hardiness, resourcefulness, and optimism [21]. This study was conducted on the sustainability and resilience of MSMEs during uncertainty for several reasons. First, previous studies recommended examining resilience because the concept differs from resilience in large businesses [22], which recommended studies on resilience because they survive in a turbulent environment. MSMEs continue to innovate, offer new knowledge, and create jobs without resources, skills, and time [23]. Second, this study included micro-enterprises as the object due to the limited literature on their resilience [24,25] recommended examining the factors influencing the failure and success or supporting micro-enterprises sustainability and their relationship with the business environment.

Third, [19] recommended examining MSMEs resilience in developing countries due to limited literature. Fourth, this study aimed to formulate policies for regency governments in Indonesia. The authors of [26] recommended examining development policies and programs at all government levels. This is because this country supports MSMEs development, but the tiered government systems with their respective responsibilities and rules are highly dynamic. Moreover, there is a need for studies on the overlapping opportunities, policies, and activities that result in inefficiency. Fifth, this study was conducted in Takalar Regency, which consists of villages. The location is essential because urban and rural areas have unique characteristics, performance, and strategies [27]. Studies on business scales and locations of micro, small, and medium enterprises were recommended by [28].

A holistic approach in the MSMEs resilience system model is needed to deal with disruptions during uncertainty [29]. This approach is needed to develop robust helpful resilience in minimizing risk, maximizing opportunity utilization, and as a role model and reference for other organizations [30]. The model designed is insufficient to rely on resources as traditional resilience sources. This necessitates a review of implementing entrepreneurial behavior as a holistic approach [11,31]. Entrepreneurship is an essential factor contributing to corporate success [32].

The company's performance is maintained while facing disruption when enough resources are available [33]. Financial resources are the most critical factor for small com-

panies dealing with dynamic situations [34]. Lack of resources, specifically budget and low skills, significantly affect company resilience [24]. Therefore, flexibility and the ability to reconfigure activities, capabilities, and resources are needed when facing unexpected disruptions during uncertainty [35]. Business success is supported by entrepreneurial orientation, which includes innovativeness, proactiveness, risk-taking, competitive aggressiveness, and autonomy [36]. Configuration of entrepreneurial orientation application and high access to capital in a dynamic environment increase business performance [34].

Corporate entrepreneurship consists of structure, strategy, and entrepreneurial process [37]. Small companies with a simple structure respond swiftly to changes in the market and recognize opportunities quickly. Businesses always look for challenging strategies to cope with an unpredictable competitive environment [38]. The ability to process change management is critical to creating resilience. This implies driving internal resources to implement change management and attention to long-term planning and communication with external parties [39].

The entrepreneurship ecosystem comprises culture, policy, finance, human resources, markets, and support. Companies that pay attention to their members, community, organization, culture, mentors, and guides become more solid and resilient [40,41]. Attention to this ecosystem forms business resilience that maintains healthy, adaptive, and integrated functions [41]. Consequently, companies generate higher sales growth, have lower financial volatility, and have a higher chance of survival [42]. The natural resources in Takalar are in the form of land and water, and the availability of factors of production supported by the availability of markets and government support in the form of counseling, agricultural machinery assistance, and other production facilities will increase regional production. The fisheries sector is one of three sectors, namely, agriculture, forestry, and fisheries, which significantly contribute to GRDP despite experiencing many disturbances in an era of uncertainty [43–45]. The context aspect is also a potential novelty, namely, the business's scale, and the research's location. This research takes the context of resilience on the scale of micro, small, and medium enterprises located in developing countries at the district government level. More specifically, these MSMEs are located in rural areas. The scope of this research includes formulating internal and external factors that shape MSME resilience and determining MSME businesses that can withstand disturbances in an era of uncertainty by formulating strategic priorities for the Takalar Regency MSME resilience system policy in an era of uncertainty.

The fishery business is developing in six of the ten sub-districts in Takalar Regency. Opportunities to increase production in the fisheries sector in these six sub-districts can still be increased because production in the fisheries sector still needs to be improved compared to its potential. Table 1 shows the potential and realization of the production of superior commodities in the fisheries sector of Takalar Regency. The gap between the potential and realization of fishery production is enormous, and products that are still in the form of raw materials should be able to encourage the growth and development of MSMEs in Takalar Regency. However, the facts show that MSMEs mortality in Takalar Regency, including MSMEs in the fisheries sector, is very high.

**Table 1.** Potential and production realization of superior commodities in the fisheries. sector of Takalar Regency.

| Commodity | Potency (ton/tahun) | Production Realization per Year (tons) | | |
|---|---|---|---|---|
| | | **2018** | **2019** | **2020** |
| Shrimp | 9.082 | 1.908 | 1.107 | 1.550 |
| Seaweed | 2399.546 | 538.680 | 455.198 | 409.117 |
| Fish | 11.244 | 2.262 | 2.262 | 2.353 |

Source: Data from the Department of Fisheries and Maritime Affairs of the Regency Takalar Year 2019–2021.

## 2. Materials and Methods

This study used the Fuzzy Analytical Hierarchy Process (FAHP) method developed from traditional AHP, where pairwise comparisons use a fuzzy scale [46]. AHP is a multi-criteria decision-making approach where factors are arranged in a hierarchical structure [47]. The approach disintegrates something complex into simple and is used in many fields and situations [48]. The basic AHP procedure breaks down the problem into a hierarchy of various levels, makes pairwise comparisons, and assigns priorities among the elements in each hierarchy. The results are synthesized to rank alternatives by considering the element priorities at the previous level and evaluating the ratings' consistency [46]. In line with this, [49] used the Fuzzy AHP method to test the components of supply chain resilience, which consists of the following steps: (a) defining the problem and determining the desired solution, (b) screening criteria, (c) formulating the problem into a hierarchical structure, (d) forming a pairwise comparison matrix, (e) consistency test, and (f) criteria and alternatives weighting using fuzzy synthetic extents.

The FAHP scale, expressed as a 'crisp' number, is thought to be less capable of dealing with ambiguity. This study was conducted at several tax offices in Indonesia. The analysis involved mapping opinions and practical input from experts and practitioners regarding effective sustainability leadership implementation. This study used the Fuzzy Analytical Hierarchy Process (FAHP), which is helpful in selecting an alternative problem by combining the fuzzy theory and hierarchical structure analysis [50]). The method allows decision-makers to include qualitative and quantitative data in the decision model. Consequently, the decision-making for a range assessment is more convincing than in a particular value. FAHP combines AHP and a fuzzy approach. A new approach with FAHP in the analysis method using a triangular fuzzy number (TFN) as a pairwise comparison scale (Table 2). TFN is a fuzzy set theory that helps measure human subjective judgments using language. Therefore, the essence of FAHP lies in pairwise comparisons described by a ratio scale associated with the fuzzy scale [50]. Primary data were obtained through in-depth interviews and set questionnaires with twenty 20 experts, such as two government parties, a structural official, an extension worker, five banking leaders, two academics, a community development, and a village assistant. Other parties were nine entrepreneurs, consisting of five micro, two small, and two medium entrepreneurs. The size number of experts selected as respondents is not a guarantee of the validity and consistency of the results. However, it can be limited by the level of formal education, from Master's to Doctoral degrees, in the field being studied and experience in that field or coming from practitioners who work in that field even though their education is not very high. However, they may have received objective recognition for their professionalism, high performance or productivity, and good track record [50].

**Table 2.** Triangular fuzzy Number.

| Description | TFN |
|:---:|:---:|
| Absolute | (7/2, 4, 9/2) |
| Very Strong | (5/2, 3, 7/2) |
| Fairly Strong | (3/2, 2, 5/2) |
| Weak | (2/3, 1, 3/2) |
| Equal | (1, 1, 1) |

Before collecting data, a conceptual model for a decision issue must be created. The foundation of a hierarchy is the notion that entities may be split into different sets, with the entities of one group influencing the entities of other groups [47]. A hierarchical framework must be built to evaluate higher education skills and the ecosystem before selecting a transformation method. As seen in Figure 1, the central component of the Fuzzy AHP's qualitative component serves as the basis for all of the criteria for the overall goal. In order to develop crucial criteria for the MSME in Takalar Regency and conduct a prior strategy to improve sustainability, the pertinent literature was researched. As previously stated, the

first level of the assessment framework states that a strategic change decision's targeted purpose is to improve the sustainability of MSMEs.

As illustrated in Figure 1, the construction of the hierarchy begins with selecting the priority strategy to create a strategy policy for sustainability MSME in Takalar Regency. After that, the factors, actors, objectives, and alternative strategies based on focus group discussion and in-depth interviews to validate the hierarchy structure are determined. The first hierarchy is the factors, which are: (a) entrepreneurial orientation, (b) corporate entrepreneurship, (c) entrepreneurship ecosystem, and (d) resources. The second hierarchy is the actors who plays a role in creating effective sustainability MSMEs, which are: (a) entrepreneurs, (b) MSMEs, (c) government, (d) higher education, (e) communities, (f) financial institutions, (g) families, and (h) associations. The third hierarchy is the objectives to be achieved, which are: (a) MSMEs have high resilience, (b) entrepreneurs have resilience, (c) acceptable market, (d) government policy supportive and (e) managed natural resources. The fourth hierarchy is an alternative strategy that can be carried out: (a) digitalization, (b) social entrepreneurship-based business, and (c) innovative transformation. The hierarchical model is as follows.

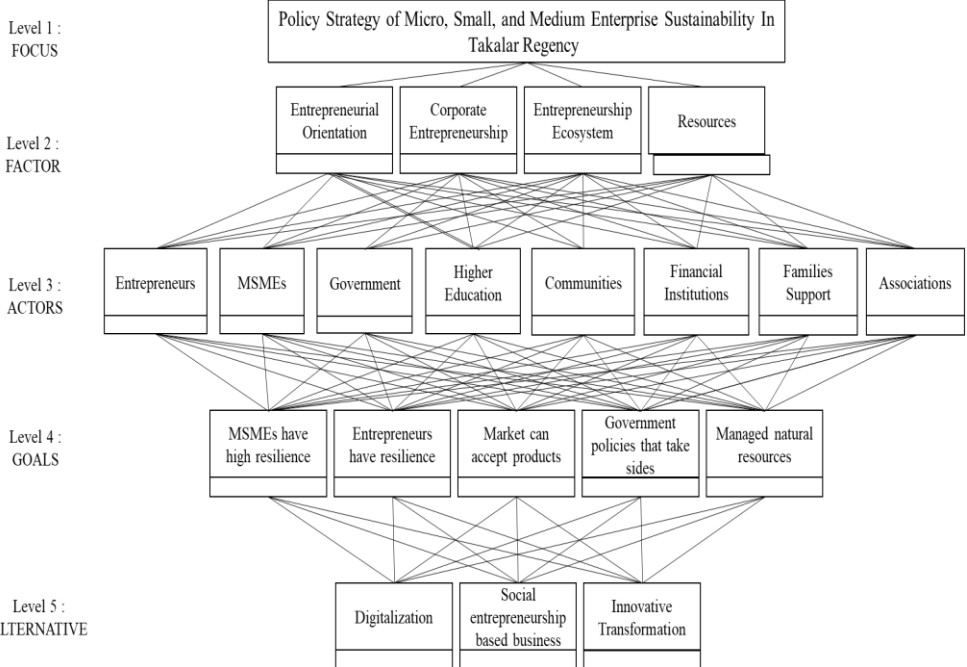

**Figure 1.** Fuzzy Analytical hierarchy model.

## 3. Results

Fuzzy AHP analysis showed the focus, factors, actors, objectives, alternative strategies, and their weights. Table 3 shows that the main factor in Fuzzy AHP analysis that can be used in designing the MSME sustainability policy strategies is the Entrepreneurship Orientation (0.256). The second factor is the Entrepreneurship Ecosystem (0.252), the third factor is Resources (0.249), and the fourth factor is Enterprise Entrepreneurship (0.243). The position of the entrepreneurial orientation factor as the dominant factor is not eliminate the role of the other three factors. The dominance of the entrepreneurial orientation factor is in line with the results of the descriptive analysis, which shows that the entrepreneurial orientation factor, especially the innovation dimension, has been used by SMEs in the fisheries sector of Takalar Regency to survive in the face of disturbances and are needed to develop.

**Table 3.** Factor weight on the factor to sustainable MSME.

| No. | Factor | Priority | Priority |
|---|---|---|---|
| 1 | Entrepreneurial Orientation | 0.256 | 1 |
| 2 | Corporate Entrepreneurship | 0.243 | 4 |
| 3 | Entrepreneurship Ecosystem | 0.252 | 2 |
| 4 | Resources | 0.249 | 3 |

All actors have yet to implement the entrepreneurial orientation factor in supporting the establishment of MSME resilience in Takalar Regency. Therefore, the sustainability system strategy that has been formulated will encourage the application of the entrepreneurial orientation factor by all actors. One of the results obtained from applying entrepreneurial orientation to entrepreneurs is increased entrepreneurial competence. Several findings have emphasized that entrepreneurial competence is one of the sources of MSME sustainability [51,52].

Based on the Fuzzy AHP analysis, the weight of each actor in Table 4. is obtained, namely, SMEs (0.132), entrepreneurs (0.1262), government (0.1260), financial institutions (0.125), families support (0.124), community (0.1235), associations (0.123), and higher education (0.12). The highest weight in this actor analysis is MSMEs, so MSMEs are positioned as priority actors that must be considered in designing MSME sustainability policies strategies. The second factor is entrepreneurs with a weight of 0.1262; strong entrepreneurs can be formed by increasing their entrepreneurial abilities through training activities, mentoring, being active in entrepreneurial activities, and connecting with networks or associations so that they can share experiences in building a business [53]. Based on the focus group discussion analysis, it is stated that several problems still need to be solved by MSMEs. Therefore, entrepreneurs who have entrepreneurial competencies are expected to find solutions. The third factor is the government, with a weight of 0.1260; MSMEs need government support in all aspects. Government support is expected to strengthen MSMEs because MSMEs that receive government support have a greater chance of being sustainable [54].

**Table 4.** Actor weight on the Policy strategy to improve sustainable MSME's.

| No. | Actor | Priority | Priority |
|---|---|---|---|
| 1 | Entrepreneurs | 0.1262 | 2 |
| 2 | MSME's | 0.132 | 1 |
| 3 | Government | 0.126 | 3 |
| 4 | Higher Education | 0.120 | 8 |
| 5 | Communities | 0.1236 | 6 |
| 6 | Financial Institutions | 0.125 | 4 |
| 7 | Families support | 0.124 | 5 |
| 8 | Association | 0.123 | 7 |

The objectives that must be built on the MSME sustainability strategy policy in Takalar Regency in Table 5. sequentially according to the weights obtained from the Fuzzy AHP analysis are: the market can accept products (0.2012), MSMEs that have high resilience (0.2011), government policies that take sides (0.2), managed natural resources (0.2), and entrepreneurs who have resilience (0.197). Based on this weight, the market goal of being able to accept MSME products is a top priority in the MSME resilience system policy for the fisheries sector in the Takalar district. This was also strengthening the market is one of the weaknesses of MSMEs. This follows the previous study of [8,10], which emphasizes the importance of paying attention to the market because neglecting the market will be a source of business failure.

**Table 5.** Objectives for Policy sustainable MSME's.

| No. | Objectives | Priority | Priority |
|:---:|:---:|:---:|:---:|
| 1 | MSME's have high resilience | 0.2011 | 2 |
| 2 | Entrepreneurs have resilience | 0.197 | 5 |
| 3 | Market can accept products | 0.2012 | 1 |
| 4 | Government policies that take sides | 0.2 | 3 |
| 5 | Managed natural Resources | 0.2 | 4 |

The focus group discussion results show that the problems experienced by MSMEs are limited supply and some fishery commodities produced by Takalar MSMEs do not meet market standards. Both analyzes show weaknesses in the quantity and quality aspects of the products offered by Takalar SMEs. Aspects of quantity and quality are conditions that must be met in order to be accepted by the market. Therefore, the MSME sustainable policy strategies strengthen these two aspects.

Furthermore, FAHP analysis of the three strategies produces weights with a minimal difference in value between the choices of one strategy and another. However, the highest value is recommended as the main priority. Based on this weight, the sequence of strategies is an innovative transformation with a weight of 0.338 as a priority choice. The second alternative strategy is digitization, with a weight of 0.334, and the third alternative is a social entrepreneurship-based business, with a weight of 0.329. The weight of each alternative strategy is shown in Table 6.

**Table 6.** Alternative Policy strategies for sustainable MSME's.

| No. | Alternative Strategy | Priority | Priority |
|:---:|:---:|:---:|:---:|
| 1 | Digitalization | 0.334 | 2 |
| 2 | Social Entrepreneurship based business | 0.329 | 3 |
| 3 | Innovative transformation | 0.338 | 1 |

Innovation includes efforts to create new products and or services, new processes, and business models [55]. In the descriptive analysis and focus group discussion analysis, it was found that the MSMEs in the fisheries sector in Takalar Regency have practiced innovation but have not been maximized because resources are placed as the main factor supporting the resilience of MSMEs. The need for innovation has been found in AHP's Fuzzy Analysis, which places entrepreneurial orientation as the main factor that must be paid attention to in developing a policy model for the MSME resilience system in the fisheries sector in Takalar Regency.

The results of interviews with micro-entrepreneurs, small entrepreneurs, and medium entrepreneurs show that the culture of innovation at the MSME level still needs to improve. Evidence of the low level of innovation at the MSME level is shown by manual practices in the cultivation, processing, and marketing subsystems. Therefore, the resulting MSME products have low competitiveness. This finding follows [8] that MSMEs find it challenging to face trade liberalization and want high-quality and modern products. Innovation is believed to be beneficial for MSMEs and has been proven by various studies, namely, competitive resources [56], which will experience an increasing growth trend [57], resulting in a higher level of profit than other companies [58], thus supporting business sustainability [59].

Innovative transformation strategies will continue to run slowly if the expected MSMEs take the initiative to start. This is because MSMEs have many limitations and have a view that places resources as the main factor. This strategy can work if the government is involved in its implementation. According to [60], the government's presence is needed, especially in ensuring that MSMEs are highly committed to adopting innovation [60]. Innovative transformation strategies as the leading choice must be included in the policy model of the MSME resilience system in the fisheries sector in Takalar Regency.

## 4. Discussion

Fuzzy AHP analysis showed that the main factor for designing the sustainable system policy strategies is Entrepreneurial Orientation (0.256), followed by Entrepreneurship Ecosystem (0.252), Resources (0.249), and Corporate Entrepreneurship (0.243). Although entrepreneurial orientation is dominant, the resilience policy must also consider other factors. The analysis results support [8], which stated that all factors supporting MSMEs' sustainability should be considered to avoid failure.

Applying entrepreneurial orientation increases the competence of MSMEs actors [51]. These competencies help create competitive advantages, promote company growth, and positively affect performance [61], ensuring business sustainability [52]. Therefore, [53] suggested that enterprises increase knowledge with formal and non-formal education and training and participate in business associations to increase competence.

Fuzzy AHP analysis showed that each actor's weights are 0.132, 0.1262, 0.1260, 0.125, 0.124, 0.1235, 0.123, and 0.12 for MSMEs, entrepreneurs, the government, financial institutions, families, communities, associations or groups, and college, respectively. Entrepreneurial orientation, corporate entrepreneurship, resources, and entrepreneurship ecosystem show that micro, small, and medium enterprises must be considered in designing resilience system policies. It shows that applying these four factors is essential for MSMEs to become resilient business organizations in the face of disruption during uncertainty. This is because they are dominated by self-employment patterns and traditional business organizations that hardly consider the four factors. Consequently, they become a vulnerable business group in an uncertain environment [8,11].

MSMEs survive because they produce consumptive goods and services the community needs and utilize local human resources, capital, raw materials, and production equipment. They rely only on capital from the owner's wealth but fail due to tight competition and disruptions [58]. Therefore, there is a need for a survival strategy that considers internal and environmental conditions. The stable internal condition makes them resilient to disruption [62]. The internal strengths are flexibility, preparedness, responsiveness, adaptability, and willingness to learn. Additionally, a communication strategy should be developed to mediate MSMEs with all stakeholders to support resilience [7,62].

Fuzzy AHP analysis showed the goals that must be built on the policy model of the resilience system in Takalar Regency during uncertainty. According to the analysis, the weights obtained are the market's acceptance of products (0.2012), MSME's need for high resilience (0.2011), government policies that take sides (0.2), managed natural resources (0.2), and resilient entrepreneurs (0.197). Subsequently, the market goal of accepting products is a top priority in the policy model of the resilience system in the fisheries sector in the regency. Focus on the fishery product market aims to maintain sustainability because neglecting the market results in business failure [8].

The goal of focusing on the market could be realized by preparing competitive products [10], expanding market segments, and increasing margins. This could be achieved by growing motivation, considering employee welfare, providing the best customer service, and conducting market intelligence. Since market control is a weakness, the policy of a resilience system must produce highly competitive products through creative ideas and technology [10].

The Innovative Transformation Strategy, with a weight of 0.338, was selected as the main priority. Innovative transformation means innovation-based change, including new products, processes, and business models [55]. This strategy is a solution for MSMEs with limited quality human resources, products, and marketing, simple and traditional organizations, and managerial weaknesses [7,24,63,64]. Furthermore, the strategy helps deal with trade liberalization, technological change, increased demand for high product quality, and challenging situations such as the COVID-19 pandemic that resulted in new regular policies [8,65]. Innovative MSMEs become highly competitive [56,58,66]) and realize high sales, profits, growth, and sustainability ([58,59,67,68]).

MSMEs need transformation in organization, processes, products, and marketing [69,70]. Organizational transformation involves becoming more flexible to face disruption during uncertainty [71]. This requires innovation to increase productivity by changing the manual process to a technology application [8]. The process is determined by the ability to absorb external knowledge to complement the weak internal innovation capability [71]. Additionally, transformation in product and marketing is needed to turn low-quality into highly competitive products acceptable to the broader market ([64,72]).

MSMEs' low ability to conduct innovative transformation strategies necessitates government involvement [73]. This would ensure a high commitment to adopting innovation [60] and building collaboration between micro, small, and medium companies, specifically in product innovation [74]. The micro, small, and medium enterprises that receive government policy support have more opportunities for innovation and growth [54,75].

## 5. Managerial Implication

Managerial implications for the practice: First, it can improve the quality of human resources through entrepreneurship education programs and mentoring for MSMEs and apprenticeships for prospective entrepreneurs. Entrepreneurship education, mentoring, and internships cover production techniques, marketing, business management, and entrepreneurship in general. This program can involve newly established or existing MSMEs, associations, and universities. In addition to innovation, the local government works with related stakeholders to encourage innovation in the form of production technology, product processing and digitization to produce products according to market needs, have competitiveness, and stimulate the formation of new MSMEs and increase the MSME business scale. Stakeholders that can be involved include universities, business associations, and financial institutions. Furthermore, the local government facilitates MSMEs to obtain business licenses quickly through communication strategies across related agencies and applying internet-based information technology in processing permits. Lastly, the development of supporting infrastructure for the government, independently or in collaboration with the private sector, is expected to provide.

This study combines resource theory with three entrepreneurial theories, namely, the theory of entrepreneurial orientation, corporate entrepreneurship, and entrepreneurial ecosystems. This study recommends three policy strategies. Based on these weights, the strategy sequence is innovative transformation as a priority choice. The second alternative strategy is digitization, and the third alternative is social entrepreneurship-based businesses. It has been found that the SMEs in the fisheries sector in Takalar Regency have practiced innovation but have yet to be maximized because resources are placed as the main factor supporting SMEs' resilience. The need for innovation places entrepreneurial orientation as the main factor that pays great attention to in developing a policy model for the MSME resilience system for the fisheries sector in Takalar Regency. The use of digitalization in the cultivation, processing, and marketing sub-systems will have an impact on increasing efficiency and income. This strategy can work if the government is involved in its implementation. Innovative transformation strategies as the primary option must be included in the MSME resilience system policy model in the Takalar Regency fishery sector. Moreover, applying social entrepreneurship will bring out community support for the sustainability of MSMEs. The real contribution of entrepreneurs and MSMEs to improving people's welfare.

## 6. Conclusions

A business plan facing disruption must include MSMEs' resilience during uncertainty. However, government policy assistance is required because they cannot build their resilience system independently. This study aimed to develop a policy model for the resilience system of micro, small, and medium enterprises in Takalar Regency consisting of factors, actors, goals, and alternative strategies. The main factor is entrepreneurial orientation, while the actor is MSMEs with high resilience. Additionally, the goal is the market's ac-

ceptance of the product, while innovation transformation is the alternative strategy. This study recommended managerial implications for the Takalar Regency government and stakeholders to build MSMEs' resilience during uncertainty. The implications include improving human resource quality through education and training. Moreover, there is a need for assistance in implementing entrepreneurial orientation dimensions, including innovativeness, proactiveness, risk-taking, and competitive aggressiveness. MSMEs should be facilitated to easily access credit from financial institutions and encourage adopting innovations to produce highly competitive products.

**Author Contributions:** S.S.: conceptualization, methodology, software, validation, formal analysis. M.R.: investigation, resources, data curation, writing—original draft preparation. I.A. and A.P.: writing review and editing, visualization, supervision. All authors have read and agreed to the published version of the manuscript.

**Funding:** This research received no external funding.

**Institutional Review Board Statement:** The study was conducted in accordance with the Declaration of Helsinki, and approved by the Institutional Review Board (or Ethics Committee) of NAME OF INSTITUTE (protocol code No.: 0183/STEI/WK1-S1-MJN/X/2022 and 27 October 2022).

**Informed Consent Statement:** Not applicable.

**Data Availability Statement:** Not applicable.

**Conflicts of Interest:** The authors declare no conflict of interest.

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
