# Peer review of "Selecting Priority Policy Strategies for Sustainability of Micro, Small, and Medium Enterprises in Takalar Regency"

_sustainability, doi:10.3390/su142315791_

Round 1

Reviewer 1 Report

I appreciate the aim of this research, but I didn’t find it very interesting while reading the manuscript. In my opinion, it was a feeble approach to justify the research. My observations are as follows:

1.     I find the literature very weak in filling the niche gap to develop a strategic policy for the micro, small and medium enterprises (MSMEs) resilience system to deal with disruption during uncertainty and maintain sustainability.

2.     The use of English must be improved.

3.     Some references come without DOI numbers. In the current trend, a manuscript should include the DOI number in every possible reference.

4.     I didn’t find any significant contribution to the manuscript, which is a must for any research work. A valid and reasonable contribution, please!!!

Author Response

Dear reviewer 1

First of all, we are very grateful for the willingness of the reviewers to provide some important notes that we consider very helpful in the process of improving our manuscript. We respond to every comment and revise several parts based on reviewers' suggestions (please see the attachment).  Hopefully, the revisions we make are in accordance with the reviewer's expectation

Regards

The authors

Reviewer 2 Report

The paper is interesting and it provides relevant insights for MSME to deal with uncertainty and build resilience.

However, here are some suggestion to improve:

You claim that “However, MSMEs in this country 34 are dominated by self-employment and traditional enterprises, have low productivity and 35 product quality because they use a manual system and less technology, and generally 36 serve small and local markets.”,,, but you haven’t yet mentioned which is the context of your study. So please firstly explain this. And then explain Which are the reasons for the context of the study as well as the specificities of the context considered?

Explain why Fisheries MSMEs are chosen as the sector of the study?

Why previous studies are considered limited? what are the problems related to “using descriptive qualitative analysis, design thinking, and a lean business model canvas that supports food security “ please explain it better.

Provide indication of the criteria used for choosing the 20 experts involved in the study.

Control the language and the punctuation in many cases all over the text.

Explain better the implications for theory and practice.

Provide a description of study limitations and future research area.

Some additional references useful:

Ndou, V., Mele, G., Hysa, E., & Manta, O. (2022). Exploiting Technology to Deal with the COVID-19 Challenges in Travel & Tourism: A Bibliometric Analysis. Sustainability14(10), 5917.

Oketch, J. O. (2022). Dynamic Scope of Entrepreneurial Ecosystem, SME Resilience and the future of Business post-COVID-19 in Africa. Journal of Entrepreneurship & Project Management6(1).

TANEO, S. Y. M., NOYA, S., MELANY, M., & SETIYATI, E. A. (2022). The Role of Local Government in Improving Resilience and Performance of Small and Medium-Sized Enterprises in Indonesia. The Journal of Asian Finance, Economics and Business9(3), 245-256.

Author Response

Dear reviewer 2

First of all, we are very grateful for the willingness of the reviewers to provide some important notes that we consider very helpful in the process of improving our manuscript (Please see attachment). We respond to every comment and revise several parts based on reviewers' suggestions.  Hopefully, the revisions we make are in accordance with the reviewer's expectation

Regards

The authors

Round 2

Reviewer 1 Report

Thank you for addressing the concerns. However, there is always scope for improvement. Good Luck!